# Aggregating Capacity in FL through Successive Layer Training for Computationally-Constrained Devices

**Kilian Pfeiffer**
Karlsruhe Institute of Technology
Karlsruhe, Germany
`kilian.pfeiffer@kit.edu`

**Ramin Khalili**
Huawei Research Center Munich
Munich, Germany
`ramin.khalili@huawei.com`

**Jörg Henkel**
Karlsruhe Institute of Technology
Karlsruhe, Germany
`henkel@kit.edu`

## Abstract

Federated learning (FL) is usually performed on resource-constrained edge devices, e.g., with limited memory for the computation. If the required memory to train a model exceeds this limit, the device will be excluded from the training. This can lead to a lower accuracy as valuable data and computation resources are excluded from training, also causing bias and unfairness. The FL training process should be adjusted to such constraints. The state-of-the-art techniques propose training subsets of the FL model at constrained devices, reducing their resource requirements for training. However, these techniques largely limit the co-adaptation among parameters of the model and are highly inefficient, as we show: it is actually better to train a smaller (less accurate) model by the system where all the devices can train the model end-to-end than applying such techniques. We propose a new method that enables successive freezing and training of the parameters of the FL model at devices, reducing the training's resource requirements at the devices while still allowing enough co-adaptation between parameters. We show through extensive experimental evaluation that our technique greatly improves the accuracy of the trained model (by $52.4$ p.p.) compared with the state of the art, efficiently aggregating the computation capacity available on distributed devices.

## 1   Introduction

Federated learning (FL) has achieved impressive results in many domains and is proposed for several use cases, such as healthcare, transportation, and robotics [1, 2, 3, 4, 5, 6]. As data in FL is not processed centrally but usually on *resource-constrained* edge devices, training machine learning (ML) models impose a large computational burden on these devices [7]. Additionally, FL requires communication, specifically exchanging ML model parameters from the devices to a centralized entity for aggregation. Extensive research has been done to lower the communication overhead required for FL, e.g., on the use of quantization in the communication [8, 9] or sketched updates [10]. Similarly, techniques such as partial updates [11], asynchronous aggregation [12, 13], and tier-based aggregation [14, 15] have been proposed to lower and account for varying computational throughput. While constrained computation throughput and communication capabilities can slow down FL convergence, high memory requirements for training that are imposed on devices can exclude devices completely from the FL system. This is, for example, the case in *Google GBoard* [16], where devices that do not have 2GB of memory for training are removed. Excluding devices from training

37th Conference on Neural Information Processing Systems (NeurIPS 2023).

lowers the reachable accuracy, as fewer devices participate in the training, also resulting in bias and unfairness [17].

Several techniques have been proposed to tackle these constraints, where the main idea is to train a lower complexity *submodel* on the devices and embed the trained submodel into the full higher-capacity server model. A submodel is typically created by *scaling the width* of the neural network (NN), e.g., using a subset of convolutional filters per NN layer. There exist several variations of the technique [9, 18, 19, 20]. In particular, Caldas *et al.* [9] propose *Federated Dropout (FD)*, which randomly, per round and per device, selects NN filters that are trained. Alam *et al.* [20] propose *FedRolex*, a sliding window approach, where all devices train the same submodel, and in each FL round, the used filter indices are shifted by one. While both these techniques allow training within given memory constraints, our results show (Fig. 1) that they perform worse than a straightforward baseline, i.e., using a *smaller* NN model that can be trained by all devices end-to-end. We evaluate CIFAR10, FEMNIST, and TinyImageNet in an FL setting using ResNet and scale the width of the NN down s.t. we achieve a $2 - 8\times$ reduction in training memory. We observe that training the same small model at all devices outperforms FedRolex and FD w.r.t to the final accuracy and convergence speed (we expect similar results for other subset-derived techniques), especially when enforcing a $4\times$ and $8\times$ memory reduction (as also evaluated in [20]).

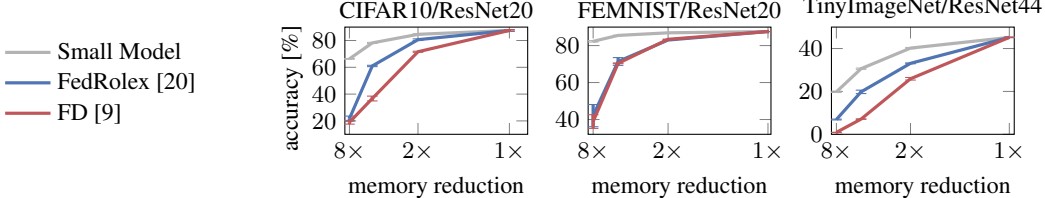

Figure 1: Accuracy of FedRolex [20] and FD [9] compared to a small model using different NN topologies and datasets after $2500/1000$ FL rounds. In the case of $1\times$, both techniques are equivalent to vanilla Federated Averaging (FedAvg). Hyperparameters of the experiments are given in Section 3.

Our results indicate that applying these techniques is rather harmful. This is as a large part of filters/parameters has to be dropped during each round of training at each device, extremely limiting the co-adaptation between parameters. Hence, the gradients for the subset of parameters that are trained on devices are calculated without considering the error of the parameters that reside on the server (more details in Appendix A). Motivated by these observations, we propose a new technique that enables successive freezing and training of the parameters of the FL model at devices, reducing the training's resource requirements at the devices while allowing a higher co-adaptation between parameters. Instead of switching between subsets of the model on an FL-round basis, we train the same parameters for several rounds and successively switch to a larger model. To obey the same memory constraints as in [9, 20], we train early layers using the full width while utilizing a scaled-down NN head. We then freeze the early layers and expand the head layers' width. By freezing early layers, no activation has to be kept in memory, hence, we lower the memory footprint. But still, the error of these frozen parameters is included in the calculation of the gradient of the new subset of parameters. We apply this technique successively till all parameters of the model are trained.

In summary, we make the following novel contributions:

- We empirically show that employing current state-of-the-art techniques, FD [9] and FedRolex [20], for memory-constrained systems can actually hurt the performance.
- We propose a novel training scheme called *Successive Layer Training (SLT)*[1], which addresses the shortcomings of previous techniques by successively adding more parameters to the training, successively freezing layers, and reusing a scaled-down NN head.
- Our evaluation of common NN topologies, such as ResNet and DenseNet, shows that SLT reaches significantly higher accuracies in independent and identically distributed (iid) and non-iid CIFAR, FEMNIST, and TinyImageNet training compared to the state of the art. Also, SLT provides a much faster convergence, reducing the communication overhead to reach a certain level of accuracy by over $10\times$ compared with FD and FedRolex. The same

---

[1]The source code of SLT is available at `https://github.com/k1l1/SLT`.

**Algorithm 1:** Successive Layer Training: $w$ and $W$ label the set of all layers' parameters.

---

**Requires:** Number of rounds $R$, devices $\mathcal{C}$, number of devices per round $|\mathcal{C}^{(r)}|$,
      configurations $S_n$ $n \in [1, \ldots, N]$, that satisfy constraint $m$, init. parameters $W^{(1)}$

**Server:**

  **foreach** round $r = 1, 2, \ldots, R$ **do**

      $\mathcal{C}^{(r)} \leftarrow$ select $|\mathcal{C}^{(r)}|$ random devices out of $\mathcal{C}$

      $w^{(r)}, S^{(r)} \leftarrow$ ConfigurationSelection$(W^{(r)}, r)$

      **foreach** *device* $c \in \mathcal{C}^{(r)}$ *in parallel* **do**

         $w^{(r)}, S^{(r)}$ receive from server

         $w^{(r,c)} \leftarrow$ DeviceTraining$(w^{(r)}, S^{(r)})$

         upload $w_j^{(r,c)}$ to server $\forall j \in \{j : K_F < j \leq K\}$

      **end**

      $w_j^{(r+1)} \leftarrow \frac{1}{|\mathcal{C}^{(r)}|} \sum_{c \in \mathcal{C}^{(r)}} w_j^{(r,c)}$   //averaging of trained layers

      $W^{(r+1)} \leftarrow w^{(r+1)}$   //layers get embedded into server model

  **end**

**ConfigurationSelection** *(W, r)*:

  $S \leftarrow$ LookupTable$(r, W)$

  $w \leftarrow W$      //scaling down NN head based on configuration

  Return $w, S$

**DeviceTraining** $(w, S)$:

  freeze $w_j$ for $j \in \{0 < j \leq K_F\}$ according to $S$

  **foreach** local mini-batch $b$ **do**

     $w \leftarrow w - \eta \nabla l(w, b)$

  **end**

  Return $w$

---

      behavior can be observed w.r.t. Floating Point Operations (FLOPs), where SLT requires $10\times$ fewer operations to reach a certain level of accuracy compared with FD and FedRolex.

- We study the performance of SLT in heterogeneous settings. We show that devices with different memory constraints can make a meaningful contribution to the global model, significantly outperforming the state-of-the-art techniques.

## 2 Methodology

### 2.1 Problem Statement and Setup

We consider a synchronous cross-device FL setting, where we have one *server* and a set of *devices* $c \in \mathcal{C}$ as participants. There is a given ML model topology $F$ on the FL server that is trained in a distributed manner for $R$ rounds. Our goal is to maximize the accuracy of the model. Similar to FD and FedRolex, we assume that a fixed number of devices $|\mathcal{C}^{(r)}|$ out of $\mathcal{C}$ participate in a round $r \leq R$. All devices are constrained in memory, and thus their training must not exceed this given memory constraint $m_{\text{constraint}}$. In other terms, we assume that no participating device can train the server NN model end-to-end.

### 2.2 Successive Layer Training

The following describes our methodology of *Successive Layer Training* for convolutional neural networks (CNNs). Firstly, we rewrite $F$ such that it is the consecutive operations of $K$ *layers*, where each layer is defined as $f_k$, $k \in [1, \cdots, K]$. Each layer $f_k$ has associated server parameters $W_k$. We label a convolution, followed by batch normalization and an activation function, a layer. Similar to [9, 20], we define a *subset* $w_k$ of the layer parameters (server) $W_k$ that is *scaled* down using $s$ as

$$w_k = W_k^{s,s} \quad w_k \in \mathbb{R}^{\lfloor sP_k \rfloor \times \lfloor sM_k \rfloor} \quad W_k \in \mathbb{R}^{P_k \times M_k}, \tag{1}$$

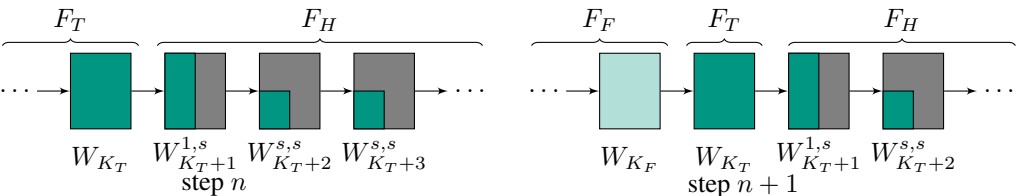

Figure 2: Visualization of SLT. With each step, $K_F$ and $K_T$ are shifted by 1. $W_1, \ldots, W_{K_F}$ denote the parameters that remain frozen during training, $W_{K_T}$ denotes parameters of layer $K_T$ that are fully trained, while $W_{K_T+1}^{1,s}, \ldots, W_K^{s,s}$ denote the parameters of the scaled-down head using $s$.

where $P_k$ labels the layer's input dimension, $M_k$ labels the output dimension of the fully-sized server parameters, and $s \in (0, 1]$ is a scaling factor (we omit the filter kernel dimensions for brevity). To obey the memory constraint on the participating devices, we split the NN into three consecutive parts. The first part of the NN contains layers that are already trained and remain frozen on the devices. The second part contains layers that are being fully trained on the devices. The last part represents the NN's *head*. To train the remaining layers within the memory budget, the head's parameters are scaled down. Throughout the FL training, we successively switch the *training configuration*, s.t., the part of the NN that is being fully trained ($s = 1$) moves from the first layer to the last layer. Thereby, successively, the remaining parameters from the scaled-down head are added. At the same time, we successively freeze more layers, starting with the first layer, to stay within the memory budget. We visualize the switching from one training configuration to the next in Fig. 2. The parts of the NN that are frozen, trained, and represent the head are labeled $F_F$, $F_T$, and $F_H$. The resulting model that is trained on the devices can be described as $F = F_F \circ F_T \circ F_H$:

- The first part $F_F$ labels the part of the NN that is frozen and where the server parameters $W_1, \ldots, W_{K_F}$ do not get updated by the devices. Freezing the first part of the NN reduces the memory overhead during training, as in the forward pass, activations do not have to be stored for frozen layers. The frozen part of the NN is defined as $F_F := \bigcirc_{k \in \{k: 0 < k \leq K_F\}} f_k$, where layers $1, \ldots, K_F$ remain frozen.
- The second part $F_T$ labels the part of the NN that is being fully trained by the devices. The parameters $W_{K_F+1}, \ldots, W_{K_T}$ get updated during training. This part is defined as $F_T := \bigcirc_{k \in \{k: K_F < k \leq K_T\}} f_k$, s.t. layers $K_F + 1, \ldots, K_T$ are fully trained.
- The last part $F_H$ describes the NN's head that is scaled down using $s$, s.t. $F_H := \bigcirc_{k \in \{k: K_T < k \leq K\}} f_k$, where the scaled-down layers $K_T + 1, \ldots, K$ are trained. The first layer of $F_H$ scales down the parameters to the width of the head s.t. $w_{K_T+1} = W_{K_T+1}^{1,s}$, where $W_{K_T+1}^{1,s} \in \mathbb{R}^{P_{K_T+1} \times \lfloor sM_{K_T+1} \rfloor}$. All consecutive layers are scaled down using $s$, s.t. $w_{K_T+j} = W_{K_T+j}^{s,s} \forall j \in [2, \ldots, K - K_T]$.

We define a set $S$ as a training *configuration* that obeys the given memory constraint, s.t. $S = \{K_F, K_T, s\}$ fully describes the mapping of layers $f_k, k \in [1, \ldots, K]$ into $F_F, F_T, F_H$, and the head's scale factor $s$. Each $S$ has a respective memory footprint during training. $m = \text{memory}(S)$ denotes the maximum memory that is utilized during training for a given configuration $S$. The maximum memory of a configuration can be determined by measurements or by calculating the size of weights, gradients, and activations that have to be kept in memory (see Section 3.2).

## 2.3 Configuration Selection

For each selected $S$, we aim to fully utilize the available memory. We define $n$ as a *configuration step* in $n \in [1, \ldots, N]$, where we add parameters to the training (i.e., *fill up* the remaining parameters of a head's layer). These steps are distributed over the total training rounds $R$ ( Fig. 3). We set $K_T = K_F + 1$, s.t. in each configuration step exactly one layer gets fully trained (filled up). We start with $K_F = K_T = 0$ (consequently $F = F_H$) to pre-train the head for a certain number of rounds. After pre-training, we increase $K_F$ by one and apply $K_T = K_F + 1$ (the first configuration has no

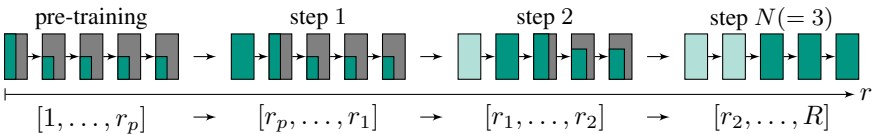

$$[1, \ldots, r_p] \quad \rightarrow \quad [r_p, \ldots, r_1] \quad \rightarrow \quad [r_1, \ldots, r_2] \quad \rightarrow \quad [r_2, \ldots, R]$$

Figure 3: Visualization of the SLT training scheme with an exemplary 5-layer NN. The model is first pre-trained for $r_p$ rounds. Following that, the model is trained for $r_1 - r_p$, $r_2 - r_1$, and $R - r_2$ rounds in configuration 1, 2, and $N(= 3)$, respectively.

frozen layers, i.e., $F = F_T \circ F_H$) and continue training[2]. We switch to the successive configuration by increasing $K_F$ by one. Hence, for the training configuration at step $n$, we have $K_F = n - 1$ and $K_T = n$, with $s_n$ selected as follows:

$$\max s_n, \quad \text{s.t.} \quad \text{memory}(S_n) \leq m_{\text{constraint}} \land s_n \leq s_j \quad \forall j \in [n+1, \ldots, N], \tag{2}$$

Equation (2) ensures that each configuration obeys the constraint $m_{\text{constraint}}$. The second constraint in Eq. (2) enforces that $s_n$ can only grow with increasing $n$ to ensure that parameters of the head are only added throughout the training but not removed. We provide a justification for maximizing $s$ instead of $F_T$ by performing an ablation study in Appendix B. The configuration selection is performed *offline*. Lastly, we define the last step $N$ where a given memory constraint (memory($S_N$)) allows for $s = 1$. If this step is reached, we train with $S_N$ for all remaining rounds since the memory budget allows to fully train all remaining parameters at once. We provide a visualization of the training process in Fig. 3 and outline SLT in Algorithm 1.

## 3 Experimental Evaluation

### 3.1 Experimental Setting and Hyperparamters

We evaluate SLT in an FL setting using PyTorch [21], where we distribute a share from the datasets CIFAR10, CIFAR100 [22], FEMNIST from the Leaf [23] benchmark, and TinyImageNet [24] to each device $c \in \mathcal{C}$, s.t. each device $c$ has a local dataset $\mathcal{D}_c$ of the same size. In each round $r$, a subset of devices $\mathcal{C}^{(r)}$ is selected. We train with the optimizer stochastic gradient descent (SGD) with momentum of 0.9, an initial learning rate of $\eta = 0.1$, and apply cosine annealing to $\eta = 0.01$ and a weight decay of $1.0 \times 10^{-5}$. We evaluate the vision models ResNet20, ResNet44 [25], and DenseNet40 [26]. For each experiment, we report the average accuracy and standard deviation of 3 independent seeds after $R$ rounds of FL training. For CIFAR10, CIFAR100, and TinyImageNet, we evaluate a scenario with $|\mathcal{C}| = 100$ devices, where each round $|\mathcal{C}^{(r)}| = 10$ devices are actively participating. For FEMNIST, we evaluate with $|\mathcal{C}| = 3550$ and $|\mathcal{C}^{(r)}| = 35$. We train for $R = 2500$ rounds for CIFAR10, CIFAR100, and TinyImageNet, and $R = 1000$ for FEMNIST. In each round $r$, each participating device iterates once over its local dataset. We apply standard image augmentation techniques like random cropping and horizontal and vertical flips to all datasets (horizontal and vertical flips are omitted for FEMNIST). An input resolution of $3 \times 32 \times 32$ is used for CIFAR and FEMNIST (up-scaled from $28 \times 28$) and $3 \times 64 \times 64$ for TinyImageNet. We use batch size 32 and perform 363 experiments in total, with an average run-time of 6 h on an NVIDIA Tesla V100.

**Comparison with the state of the art:** We compare SLT against several baselines. We introduce $I_k^{(r)}$ as the set of indices of the output dimension of a layer $k \in [1, \ldots, K]$ where $r \in [1, \ldots, R]$ denotes the rounds. Consequently, for full-sized NNs $|I_k^{(r)}| = M_k$. The subset for training is scaled down by building a dense matrix using the indices from $I_k^{(r)}$, s.t. the scaled-down parameters are $w_k \in \mathbb{R}^{\lfloor sP_k \rfloor \times \lfloor sM_k \rfloor}$. The consecutive layer's input dimension indices are equal to the last layer's output indices. The first layer's input dimension is not scaled to feed all color channels into the NN.

**Small model:** Devices train a submodel where all filters per layer are scaled down by $s$, s.t. all devices can train the submodel. The remaining filters are trained end-to-end throughout the rounds.

---

[2]We discuss in the evaluation section how to decide the number of rounds a certain configuration should be trained before switching to the next configuration.

The same submodel is used for evaluation. The indices of the output dimension of a layer $k$ are selected using $I_k^{(r)} = I_k = \{i : 0 \leq i < \lfloor sM_k \rfloor\}$.

**FedRolex [20]:** FedRolex creates a submodel by scaling the numbers of filters using $s$. Each device trains the same continuous block of indices. The filter indices trained on the devices are shifted in a rolling window fashion every round. The server averages the trained block and evaluates using the full server model ($s = 1$). The indices $I_k^{(r)}$ are selected with $\hat{r} = r \mod M_k$ using

$$I_k^{(r)} = \begin{cases} \{\hat{r}, \hat{r}+1, \ldots, r + \lfloor sM_k \rfloor - 1\} & \text{if } \hat{r} + \lfloor sM_k \rfloor \leq M_k \\ \{\hat{r}, \hat{r}+1, \ldots, M_k - 1\} \cup \{0, \ldots, \hat{r} + \lfloor sM_k \rfloor - 1 - M_k\} & \text{otherwise} \end{cases}. \quad (3)$$

**FD [9]:** FD creates a submodel by scaling down the number of filters using $s$. The indices of the filters are randomly sampled per device per round on the server. Hence, the indices $I_k^{(r,c)}$ of a device $c$ of round $r$ is a round-based per-device random selection of $\lfloor sM_k \rfloor$ indices out of all $M_k$ indices. The server aggregates the device-specific submodels after training and evaluates the full model ($s = 1$).

## 3.2 Memory Footprint during Training

The high memory requirements during training can be split into three groups: Firstly, the weights of the NN have to be stored in memory. This is required both for the forward pass and the backward pass. Secondly, for the calculated gradients in the backward pass, the activation maps of the respective layers have to be kept in memory. Lastly, the calculated gradients have to be stored in memory. In state-of-the-art CNNs, the size of the activation map makes up for most of the memory requirements, while the size of the weights only plays a minor role. For ResNet44 and DenseNet40, we measure that activations make up for $\sim 99\%$ of the required memory for training, while gradients and parameters account for the remaining $1\%$. Consequently, the required memory linearly reduces with $s$ for FD and FedRolex, as the number of layer's output channels $\lfloor sM_k \rfloor$ determines the activation map's size. Similarly, for SLT, we measure the maximum amount of memory that is required during training by counting the size of the activation maps, as well as the loaded weights and gradients in training. For frozen layers, it is only required to load the parameters in memory, while no activation maps and gradients have to be stored. For the fully trained layer $K_T$, it is required to store the layer's full parameters $w_{K_T}$, as well as the full-size activation map and gradients in memory. For all other layers (NN head), memory scales linearly with $s$. We provide implementation details in Appendix E.

We evaluate memory constraints that are given by scaling down $s$ in FedRolex and FD by $s_{\text{FD/FedRolex}} \in [0.125, 0.25, 0.5, 1.0]$ for experiments with ResNet and $[0.33, 0.66, 1.0]$ for DenseNet[3]. In SLT, for a given $s_{\text{FD/FedRolex}}$, we adjust $s_n$ for each step $n$ in the following way

$$\max s_n, \quad \text{s.t.} \quad \text{memory}(S_n) \leq \text{memory}(s_{\text{FD/FedRolex}}) \wedge s_n \leq s_j \quad \forall j \in [n+1, \ldots, N], \quad (4)$$

to ensure that our technique obeys the same constraint as the baselines. If $s_{\text{FD/FedRolex}} = 1.0$, all algorithms coincide with vanilla Federated Averaging (FedAvg) using the full server model.

We distribute the required steps $N$ over the total rounds $R$, s.t. all parameters receive sufficient training. Specifically, we distribute the rounds based on the share of parameters that are added to the training within a configuration $n$. We calculate the number of all trained parameters $Q$ by using $K_F, K_T$, and $s$ s.t.

$$Q(K_F, K_T, s) = \left( \sum_{k \in \{k: K_F < k \leq K_T\}} P_k M_k \right) + P_{K_T+1} \lfloor sM_{K_T+1} \rfloor + \sum_{k \in \{k: K_T+1 < k \leq K\}} \lfloor sP_k \rfloor \lfloor sM_k \rfloor, \quad (5)$$

and use $Q$ to calculate the share of rounds $R_n$ for a step $n$. The share of rounds for pretraining is calculated using $R_{\text{pretraining}} = R \frac{Q(0,0,s)}{Q(0,0,1)}$. For step 1, $R_1 = R \frac{Q(0,1,s)}{Q(0,0,1)} - R_{\text{pretraining}}$. For all steps $n > 1$, we calculate the rounds using

$$R_n = R \frac{Q(n-1, n, s_n) - Q(n-2, n-1, s_{n-1})}{Q(0,0,1)}. \quad (6)$$

Lastly, the switching point for pretraining is $r_{\text{pretraining}} = R_{\text{pretraining}}$ and $r_n = R \frac{Q(n-1,n,s_n)}{Q(0,0,1)}$ for all steps $n$.

---

[3]For DenseNet, SLT only enables a reduction of $3\times$, as in DenseNet, specific layers have a significantly larger sized feature map than others, which limits our technique's effectiveness w.r.t memory reduction.

Table 1: Results for iid experiments with ResNet and DenseNet using CIFAR10, FEMNIST, CIFAR100, and TinyImageNet. Accuracy in % after $R$ rounds of training is given.

| Setting | ResNet20/CIFAR10 | | | | ResNet20/FEMNIST | | | |
|---|---|---|---|---|---|---|---|---|
| $s_{FD/FedRolex}$ | 0.125 | 0.25 | 0.5 | 1.0 | 0.125 | 0.25 | 0.5 | 1.0 |
| SLT (ours) | **74.1±0.8** | **83.4±0.2** | **85.2±0.6** | 87.5±0.6 | **84.4±0.3** | **85.8±0.1** | **86.9±0.0** | 87.6±0.0 |
| Small model | 66.3±0.3 | 78.2±0.4 | 84.6±0.4 | | 82.3±0.4 | 85.5±0.1 | 86.9±0.0 | |
| FedRolex [20] | 21.7±1.9 | 61.0±0.5 | 80.6±0.6 | | 42.1±6.0 | 71.4±2.1 | 83.0±0.1 | |
| FD [9] | 19.0±1.4 | 36.7±1.8 | 71.6±0.4 | | 38.9±3.7 | 70.4±0.2 | 83.4±0.3 | |

| Setting | DenseNet40/CIFAR100 | | | ResNet44/TinyImageNet | | | |
|---|---|---|---|---|---|---|---|
| $s_{FD/FedRolex}$ | 0.33 | 0.66 | 1.0 | 0.125 | 0.25 | 0.5 | 1.0 |
| SLT (ours) | **51.1±0.4** | 53.3±0.6 | 60.2±0.5 | **33.5±0.1** | **40.3±0.5** | **42.3±0.2** | 45.2±0.1 |
| Small model | 43.9±1.5 | **55.9±0.1** | | 19.8±0.3 | 30.6±0.3 | 40.2±0.3 | |
| FedRolex [20] | 22.2±0.3 | 46.7±0.1 | | 6.9±0.2 | 19.8±0.8 | 33.1±0.2 | |
| FD [9] | 13.5±0.5 | 41.9±1.5 | | 0.9±0.0 | 7.1±0.1 | 25.9±0.6 | |

Preliminary experiments have shown that this mapping scheme outperforms other techniques, like an equal distribution of rounds to all configurations, and enables SLT to converge as fast as a small model while reaching a significantly higher final accuracy (we provide further results in Appendix C). The mapping of steps $N$ to rounds $R$ does not rely on private data (or any data) and can be stored in a look-up table prior to the training. A visualization of SLT is given in Fig. 3. We provide the number of steps $N$ for different NN architectures and constraints in Appendix C.

## 3.3 Experimental Results

**Iid results:** For the iid case, results are given in Table 1. We observe that SLT reaches significantly higher accuracy for ResNet20 and CIFAR10 for all evaluated constraints, outperforming a small model baseline by up to $7.8$ p.p. and state of the art by $52.4$ p.p.. The results with FEMNIST show that a small model baseline already provides sufficient capacity for the dataset since only a few percentage points separate $s_{FD/FedRolex} = 0.125$ from the full model (i.e., when $s_{FD/FedRolex} = 1$). Hence, SLT can only provide a minor benefit over a small model baseline. The contrary can be observed for CIFAR100 and TinyImageNet, where using a small model ($s_{FD/FedRolex} = 0.125$) loses up to $25.4$ p.p. to the full model. Additionally, it can be observed that for low memory constraints, FD and FedRolex fail to learn a useful representation at all. SLT improves upon a small model by up to $13.7$ p.p. and up to $26.6$ p.p. compared to state of the art.

**Non-iid results:** Typically, data in FL is not distributed in an iid fashion but rather non-iid. We repeat the experiments shown in Table 1 but distribute the data on the devices in a non-iid fashion. Similar to [27], we apply a Dirichlet distribution, where the rate of non-iid-ness can be varied using $\alpha$. For all experiments, we set $\alpha = 0.1$. We observe from Table 2 that the small model baselines in the case of CIFAR10 and FEMNIST lose accuracy compared to the full model. Hence, the gain of SLT compared to a small model baseline increases. The results for CIFAR100 and TinyImageNet show a proportional drop in accuracy for all algorithms. However, SLT still outperforms other techniques by a large margin. Note that we could apply common non-iid mitigation techniques like FedProx [11] on top of SLT to further limit the drop in accuracy.

For additional experimental results, we refer the readers to Appendix D.

**Communication, computation, and convergence speed:** We evaluate our technique w.r.t. the communication overhead of the distributed training and the number of computations devices have to perform (FLOPs). Specifically, we evaluate the gain in accuracy over required transmitted data and performed FLOPs. We show the results in Fig. 4. We observe that our technique converges fast, similarly to a small model, while reaching higher final accuracy. Compared to FD and FedRolex, our technique requires significantly less communication to reach the same level of accuracy. Similar behavior can be observed w.r.t. FLOPs.

## 3.4 Heterogeneous Memory Constraints

Memory constraints in devices can be heterogeneous. We evaluate SLT in such scenarios and compare it against the start of the art. We evaluate with different resource levels and split the available constraint

Table 2: Results for non-iid experiments with ResNet and DenseNet using CIFAR10, FEMNIST, CIFAR100, and TinyImageNet. Accuracy in % after $R$ rounds of training is given.

| Setting | ResNet20/CIFAR10 | | | | ResNet20/FEMNIST | | | |
|---|---|---|---|---|---|---|---|---|
| $s_{\text{FD/FedRolex}}$ | 0.125 | 0.25 | 0.5 | 1.0 | 0.125 | 0.25 | 0.5 | 1.0 |
| SLT (ours) | **52.4±0.9** | **69.6±0.6** | **75.5±1.3** | 80.5±1.3 | **81.2±1.6** | **83.0±2.0** | **83.8±1.9** | 84.0±1.9 |
| Small model | 44.7±1.2 | 63.1±0.7 | 73.6±0.6 | | 79.6±0.8 | 82.9±1.1 | 83.3±2.4 | |
| FedRolex [20] | 15.0±3.7 | 29.8±1.7 | 48.3±2.9 | | 39.4±2.0 | 59.3±2.1 | 78.5±0.5 | |
| FD [9] | 11.3±0.9 | 10.7±0.6 | 34.9±5.7 | | 15.9±8.2 | 51.0±1.2 | 79.7±1.1 | |

| Setting | DenseNet40/CIFAR100 | | | ResNet44/TinyImageNet | | | |
|---|---|---|---|---|---|---|---|
| $s_{\text{FD/FedRolex}}$ | 0.33 | 0.66 | 1.0 | 0.125 | 0.25 | 0.5 | 1.0 |
| SLT (ours) | **45.9±1.4** | 48.4±0.5 | 55.8±0.5 | **28.5±1.2** | **35.1±1.1** | **36.1±0.2** | 39.0±0.8 |
| Small model | 40.5±1.2 | **51.8±0.2** | | 16.9±0.2 | 25.3±0.5 | 34.2±0.4 | |
| FedRolex [20] | 20.0±0.3 | 42.9±0.4 | | 1.5±0.5 | 12.7±1.3 | 26.1±0.5 | |
| FD [9] | 7.6±0.1 | 36.9±1.0 | | 0.4±0.1 | 0.6±0.0 | 20.0±1.3 | |

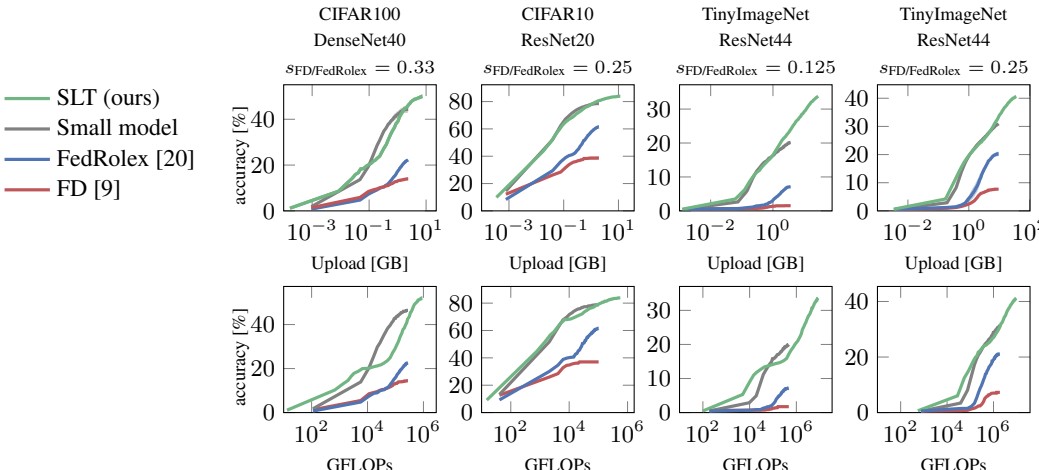

Figure 4: Maximum reached accuracy in % over data upload and performed FLOPs for CIFAR10, CIFAR100, and TinyImageNet using DenseNet40, ResNet20, and ResNet44 in an iid training case.

levels equally upon the devices, i.e., when constraints of $s_{\text{FD/FedRolex}} = [0.125, 0.25]$ are given, $50\%$ of the devices train with $s_{\text{FD/FedRolex}} = 0.125$ while the remaining $50\%$ use $s_{\text{FD/FedRolex}} = 0.25$. SLT supports heterogeneous constraints through the following mechanism: Firstly, devices with the highest constraint perform training as done in the homogeneous case, outlined in Algorithm 1 using the head's scale factor as described in Eq. (2). Devices that are less constrained use the same scale factor $s_n$ per configuration to ensure that all devices train the same number of parameters within a layer. To utilize the remaining available memory, less constrained devices freeze fewer layers, therefore, train more layers at full width. For a given $K_T$ and $s_n$ of a configuration $S_n$, the remaining memory of less constrained devices is utilized by minimizing $K_F$, s.t.

$$\min K_F, \text{ s.t. } \text{memory}(S_n) \leq \text{memory}(s_{\text{FD/FedRolex}}). \quad (7)$$

In addition to FD and FedRolex, we evaluate HeteroFL [18] and FjORD [19]. Both require that some devices are capable of training the full NN end-to-end, otherwise, some parameters do not receive any updates. In cases where no device can train the server model end-to-end, we reduce the size of the server model such that at least one participating device can fully train the model.

**Small model**: All devices train a small model regardless of their constraints. Scale factor $s$ is set to the minimum a participating device supports.

**FedRolex** [20]: Similar to the homogeneous case, FedRolex uses a rolling window (Eq. (3)). In heterogeneous cases, devices use a constraint-specific scale $s_e$, to adjust the number of filters $\lfloor s_e M_k \rfloor$.

Table 3: FL with heterogeneous constraints. Accuracy in % after $R$ rounds of training is given.

| Setting | DenseNet40/CIFAR100 | | ResNet44/TinyImageNet | | |
|---|---|---|---|---|---|
| $s_{\text{FD/FedRolex}}$ | [0.33, 0.66] | [0.33, 0.66, 1.0] | [0.125, 0.25] | [0.125, 0.25, 0.5] | [0.125, 0.25, 0.5, 1.0] |
| SLT (ours) | **46.4±2.0** | **49.3±1.8** | **30.3±1.2** | **33.0±0.5** | **35.9±0.4** |
| Small model | 40.5±1.2 | 40.5±1.2 | 16.9±0.2 | 16.9±0.2 | 16.9±0.2 |
| FedRolex [20] | 33.2±0.4 | 43.9±1.3 | 5.4±0.2 | 13.8±1.3 | 23.6±0.7 |
| FD [9] | 21.2±0.6 | 38.1±0.4 | 0.5±0.1 | 0.6±0.1 | 20.6±1.7 |
| HeteroFL [18] | 42.2±1.3 | 42.8±0.5 | 20.7±0.7 | 24.1±0.2 | 23.3±0.5 |
| FjORD [19] | 38.7±0.4 | 36.9±0.4 | 22.4±1.0 | 25.3±0.3 | 27.5±0.8 |

**FD** [9]: Although Caldas *et al.* [9] do not specifically evaluate heterogeneous devices, heterogeneity can be supported straightforwardly by using constraint-specific $s_e$ for scaling down the NN. This extension of FD is also evaluated in FedRolex and FjORD.

**HeteroFL** [18]: In HeteroFL, devices use the same subset throughout the training. To support heterogeneity, devices use a resource-specific scaling factor $s_e$, s.t. for each $s_e$ the indices are selected using $I_k^{(r,e)} = I_k^{(e)} = \{i \mid 0 \le i < \lfloor s_e M_k \rfloor\}$.

**FjORD** [19]: FjORD uses the same indices as HeteroFL, but each device switches between constraint-specific subsets that satisfy the device constraints within a local epoch on a mini-batch level.

**Heterogeneity results**: We repeat the experimental setup as presented in Table 2 for TinyImageNet and CIFAR100, but enforce varying device constraints in the experiments (see Table 3). We observe that SLT outperforms others in all evaluated scenarios. FedRolex can improve upon a small model in some settings, but this is not the case with FD. For FjORD and HeteroFL, we observe that both outperform the small model baseline. Yet, in some cases, both HeteroFL and FjORD have a lower accuracy when utilizing more constraint levels. For HeteroFL, it can be observed that using ResNet44 with 4 constraint levels reaches a lower accuracy than with 3 levels (despite the fact that all devices have higher average resources $\mathbb{E}[s_{\text{FD/FedRolex}}]$ of $\approx 0.47$ in the case of $[0.125, 0.25, 0.5, 1.0]$ instead of $\approx 0.29$ in the case of $[0.125, 0.25, 0.5]$). The same can be observed for FjORD with DenseNet40. As both techniques, in principle, use the same subset mechanism as FedRolex and FD, we think that both suffer from supporting more constraint levels that cause less co-adaptation between NN filters.

## 4 Related Work

We cover related work that studies similar problems or employs similar techniques.

**Resource constraints in FL**: Most works on resource-constrained FL target communication. Specifically, the use of quantization and compression in communication [9, 8] and sketched updates [10] have been proposed to lower the communication burden. Chen *et al.* [28] propose *adaptive parameter freezing* as they discover that parameters stabilize during training and do not have to be transferred to the server. Another branch of work focuses on reducing communication, computation, and memory requirements by employing only a subset of the full NN on devices. Caldas *et al.* [9] introduce *FD*, a mechanism that creates a device-specific subset by randomly selecting a subset of CNN filters. Diao *et al.* [18] introduce HeteroFL, which allows for heterogeneous constraints by employing fixed subsets of different sizes to the NN and aggregating them on the server. Horvath *et al.* [19] introduce a similar technique (FjORD), with the main difference that in FjORD, each device trains every available subset within its capabilities. Rapp *et al.* [29] propose DISTREAL, a technique that uses varying subsets on a mini-batch level such that devices finish their update on time despite having intra-round changing resources. FedRolex [20] supports heterogeneity similar to FjORD and HeteroFL but allows for server NN models that are outside of the capabilities of all devices. This is enabled by not training a fixed subset of the NN parameters but by training a rolling window of all parameters that is shifted on a round basis. Beyond subsets, the use of low-rank factorization [30, 31] has been proposed to train NN models. Lastly, Qui *et al.* [32] propose sparse convolutions to lower the resource requirements for training but require special hardware for sparse computations to realize the gains.

**Layer-wise model training:** Layer-wise model training has been proposed in centralized training of CNNs as an alternative to training with end-to-end backpropagation of the error. Hettinger *et al.* [33] introduced a technique that adds CNN layers one at a time during training using auxiliary

heads for classification while freezing early layers. Similar techniques have also been employed for unsupervised learning, where representations are trained with contrastive techniques without requiring end-to-end gradient propagation [34, 35]. Recently, the concept of progressive model growth has also been proposed for FL: Wang *et al.* [36] propose *ProgFed*, where they discover that by progressively adding CNN layers to the NN while using an auxiliary head, the FL training converges faster and required less communication to reach the same accuracy as an end-to-end baseline. Similarly, Kundu *et al.* [37] propose a technique that grows the model depending on the complexity of the data to reach a high accuracy if the NN capacity is not sufficient for the problem. *Importantly, both techniques only focus on increasing the convergence speed. Hence, they consider communication and computation overhead but not the problem of constrained memory on edge devices, nor do they support heterogeneous devices. In both techniques, eventually, all devices have to train the full-size NN and, consequently, have to have the memory resources available for that.*

**Memory-efficient training**: Several techniques have been proposed to train an ML model in a memory-efficient way. Kirisame *et al.* [38] present Dynamic Tensor Rematerialization that allows recomputing activation maps on the fly. Similarly, encoding and compression schemes [39, 40] have been proposed to lower the size of the activation maps during training. Techniques like that trade memory for computation, and some lower the accuracy by using approximation or lossy compression. *Importantly, these techniques are orthogonal to SLT, FD, FedRolex, HeteroFL, and FjORD.*

## 5  Conclusion

We proposed SLT that is able to reduce the memory requirements for training on devices, efficiently aggregating computation capacity and learning from all available data. Through extensive evaluation, we show that gains in final accuracy as well as the faster convergence speed (compared with state of the art) are robust throughout different datasets, data distribution, and NN topologies.

**Limitations:** We observe that SLT is most effective if the used NN architecture is deep (i.e., has many layers), as the cost of *filling up* a single layer becomes less significant. Also, SLT is less effective if the size of the activation map is strongly unevenly distributed throughout the layers (DenseNet), as it has to adapt to the layer with the highest memory requirements when filled up. Besides, we applied SLT to CNN topologies only. Finally, we mainly focused on memory as a *hard constraint* [41] for training. We show that communication and FLOP efficiency are significantly higher than in the state of the art, but we did not consider per-round communication or FLOP constraints. For future work, we want to extend our study to other topologies, such as transformers, and employ neural architecture search (NAS) techniques to find NN configurations that reach the highest accuracy when trained in FL with SLT in heterogeneous environments.

**Broader impact:** Our solution could help reduce biases in FL systems, improving fairness. For instance, by including users with low-end smartphones in the learning process, it provides these users (who perhaps cannot afford high-end devices) with better experiences as the model is going to be trained over their data, too. It could also reduce the cost of deployment of distributed IoT systems (e.g., sensor networks), as they can be implemented with low-cost devices (or a mixture of low and high-cost devices), enabling, e.g., deployment of larger and more fine-grained monitoring systems. On the negative side, distributing learning over low-end devices that are not particularly designed for training tasks can increase the overall energy consumption of the system. This is an important issue that should be studied in more detail.

## Acknowledgments and Disclosure of Funding

This work was partially funded by the "Helmholtz Pilot Program for Core Informatics (kikit)" at Karlsruhe Institute of Technology. The authors acknowledge support by the state of Baden-Württemberg through bwHPC.

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

# A Co-Adaptation in Subset-based FL

Federated Dropout is originally inspired by regular dropout [42], a regularization technique that constrains the capacity of a large NN model by randomly dropping parameters from the training, thereby limiting co-adaptation among parameters of the model. This is essential to improve the accuracy and reduce the over-fitting of parameters, as shown in various studies. FD and FedRolex adopt the dropout technique, removing CNN's filters in a round-based manner. These techniques, however, exercise dropout to its extreme, dropping a large part of filters so that not enough co-adaptation between filters remains. In particular, the gradients for the subset of parameters trained on a device are calculated without consideration of the error of the remaining parameters that reside on the server. These subsets are randomly changing over time and devices, reducing the co-adaptation of this distributed training process. Add to these the fact that the data is also distributed over devices, so applying such a random scheme significantly decreases the chance that a subset of parameters is being trained together over a sizable proportion of the data.

To further study the effects of co-adaptation on the reachable accuracy and the differences between FD and FedRolex, we run the following experiment, using CIFAR10 with ResNet20 and $s_{\text{FD/FedRolex}} = 0.25$:

- We modify FD s.t. all devices train the same random subset per round, i.e., the same indices $I^{(r,c)} = I^{(r)}$ per round (index $k$ is omitted for simplicity).
- We limit the randomness, where at each round, we arbitrarily select $I^{(r)}$ out of a set $\mathcal{I} = \{I_1, \ldots, I_{|\mathcal{I}|}\}$ of randomly initialized subsets that are generated once prior to training.

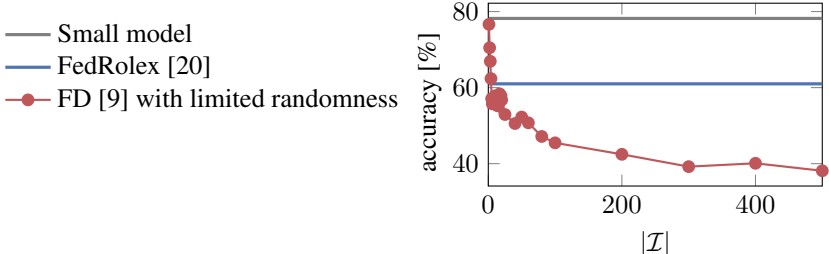

Figure 5: FD with limited randomness using CIFAR10 with ResNet20 and $s_{\text{FD/FedRolex}} = 0.25$.

We vary the randomness by varying the number of sampled subsets in $\mathcal{I}$, s.t. $|\mathcal{I}| \in [0, 500]$. Thereby, the probability of a specific subset being selected is $p = \frac{1}{|\mathcal{I}|}$. Recall that all the devices train the same submodel in a round, and there are only $|\mathcal{I}|$ submodels that would be trained by devices over the training period. Evaluation is done with the full server model. Results are shown in Fig. 5.

We observe the following effects: 1) The final accuracy drops proportionally to $\sim p$. 2) In the case of $|\mathcal{I}| = 1$, FD behaves similarly to the small model baseline, as always the same subset is used for training. We also observe that remaining untrained filters have a minor effect on the accuracy when compared with a small model. However, because of these untrained parameters, the model fails to reach higher accuracies as with SLT (see Section 3). 3) The accuracy drops with introducing more randomness to the training process (i.e., increasing $|\mathcal{I}|$). This is as co-adaptation among parameters of the model reduces as we increase the randomness. 4) The rolling window approach of FedRolex is a special case of FD with limited randomness (i.e., $|\mathcal{I}| = 5$ in this experiment).

# B Ablation study maximizing $s$ over $F_T$

To justify our design choice in Section 2 to maximize $s_n$ for all steps $n$, we conduct an ablation study, where we study the best trade-off between $s$ and $F_T$. In particular, instead of maximizing $s$, we only use fractions of the maximized $s_n$ labeled $s_{\text{ablation}}$. We evaluate different values for $s_{\text{ablation}}$, i.e. $\frac{s_{\text{ablation}}}{s_n} \in (0, 1]$. When only a fraction of the maximized $s_n$ is used in a step, the remaining memory can be used to increase the size of $F_T$. We conduct with CIFAR10/ResNet20 and TinyImageNet/ResNet44, where all the hyper-parameters are kept the same as in Section 3.1 (except the changes in $s_n$ and $F_T$). The final accuracy of SLT is displayed in Fig. 6. For each run, we depict the

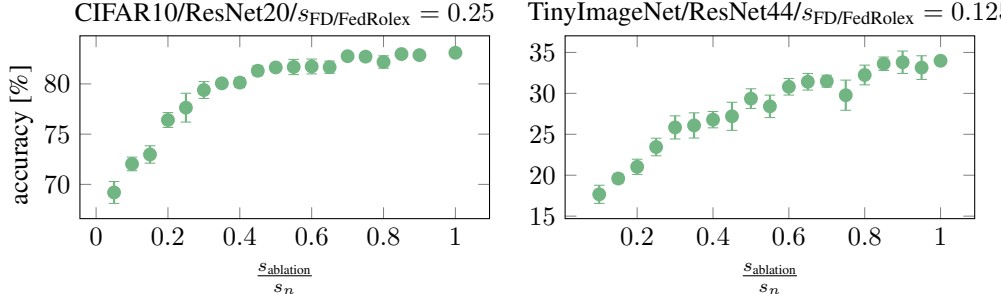

Figure 6: Trade-off between maximizing $s$ over $F_T$. The results show that maximizing $s$ ($\frac{s_{\text{ablation}}}{s_n} = 1$) gives the highest accuracy.

Table 4: Steps $N$ in SLT for different constraints $s$ and NN models.

| Constraint $s$ | ResNet20 | ResNet44 | DenseNet40 |
|---|---|---|---|
| 0.66 | - | - | 10 |
| 0.5 | 8 | 14 | - |
| 0.33 | - | - | 15 |
| 0.25 | 14 | 26 | - |
| 0.125 | 16 | 14 | - |

average accuracy and standard deviation for three seeds. The results show that by maximizing $s$ in favor of $F_T$, SLT reaches a higher final accuracy.

## C Mapping of steps $N$ to rounds $R$

Generally, any layer that is trained in SLT should receive a sufficient amount of training to extract useful features for downstream layers, but at the same time, it should not overfit in the current configuration. The mapping of rounds $R$ to steps $N$ in SLT is done proportionally to the amount of added (previously untrained) parameters to the training. Since $N$ depends on how many steps are required until $s = 1$, the number of steps depends on the constraint level. In Table 4, we list $N$ for all experiments and constraints. Fig. 7 visualizes this mapping for DenseNet40, ResNet20, and ResNet44, where SLT's accuracy over rounds is displayed in green while steps over rounds are displayed in black. This mapping scheme has key advantages over other techniques. Most importantly, it depends only on the NN structure and not on the data available on the devices. Hence, it can be calculated offline prior to the training.

We compare our mapping scheme with two other mapping schemes, one offline and one online:

- **Equal distribution:** In this scheme, we equally distribute the rounds to the steps, i.e., $R_n = \frac{R}{N+1}$.
- **Early stopping:** In this scheme, we decide online, based on the test accuracy, when to switch. If the test accuracy on the server for a number of FL rounds does not improve, the mapping switches to the next configuration. The number of rounds is usually referred to as *patience*. We evaluate with patience 5, 15, and 25.

To compare the mapping schemes, we run experiments with CIFAR10 and ResNet20, where, except for the mapping scheme, all hyperparameters are kept the same (as presented in Section 3). We can observe from the results in Fig. 8 that our mapping scheme outperforms the others with respect to the final accuracy and convergence. Even though the early-stopping-based technique with patience 5 increases $n$ more aggressively, it does not result in faster convergence or higher final accuracies.

## D Miscellaneous Experiments

To evaluate how SLT and baselines perform with more complex datasets and deeper NNs, we evaluate it with the full ImageNet [43] dataset (1.28M images, 1K classes). However, we downscale the

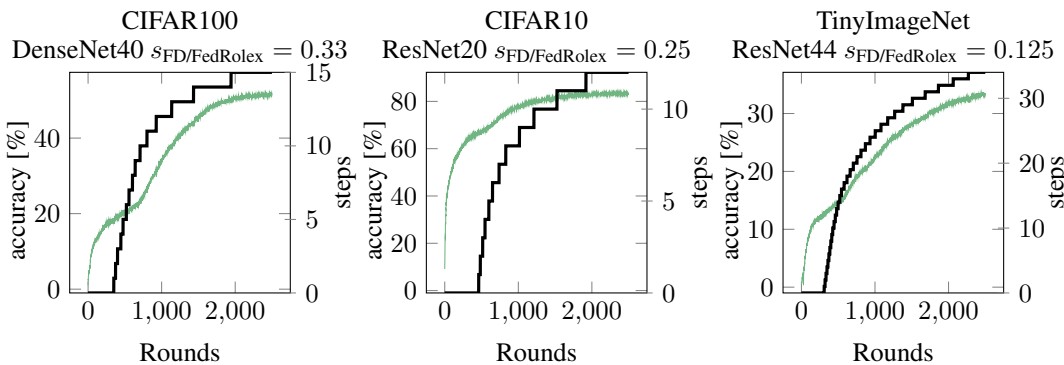

Figure 7: Accuracy over rounds (green) and steps over rounds (black) in SLT for CIFAR100/DenseNet40, CIFAR10/ResNet20, and TinyImageNet with ResNet44.

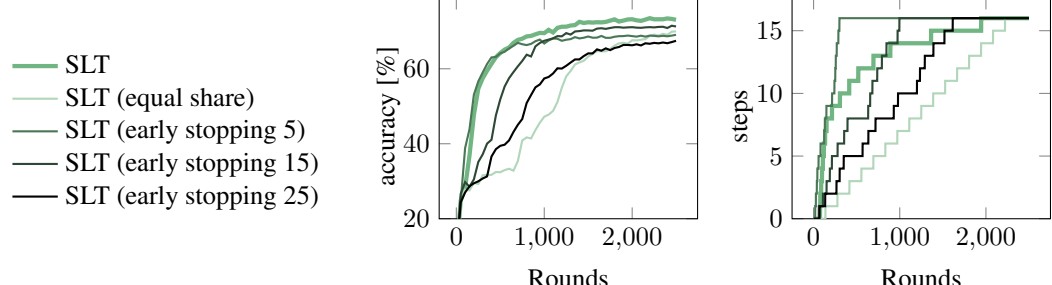

Figure 8: Different strategies for mapping $N$ to $R$ using CIFAR10 with ResNet20 and $s_{\text{FD/FedRolex}} = 0.25$. Accuracy over rounds is displayed on the left. Steps over rounds are displayed on the right.

images to $3 \times 64 \times 64$ pixels to reduce the complexity of the evaluation. To account for the larger dataset, we increase the number of devices to $|\mathcal{C}| = 500$ and the number of rounds to $R = 4000$. All remaining hyperparameters are kept the same (Section 3). We use ResNet56 to account for the more complex dataset. Results for iid and non-iid data are provided in Table 5. We observe that the general trend of Tables 1 and 2 remains the same: SLT outperforms the state of the art and the small model baseline with large margins.

## E  Training Memory Measurements in PyTorch

We measure the maximum memory requirement for the evaluated NN models ResNet and DenseNet using PyTorch 1.10. Specifically, we measure the size of the activations, gradients, and weights. These memory measurements are done offline (prior to training) and do not require any data.

- **Measurement of weights**: To measure the size of the weights, we sum up all tensors that are present in the NN's `state_dict`.

Table 5: Results for iid and non-iid experiments with ResNet56 using Imagenet ($64 \times 64$) are given. Accuracy in % after 4000 rounds of training is given.

| Setting | ResNet56/ImageNet/iid | | | | ResNet56/ImageNet/non-iid | | | |
|---|---|---|---|---|---|---|---|---|
| $s_{\text{FD/FedRolex}}$ | 0.125 | 0.25 | 0.5 | 1.0 | 0.125 | 0.25 | 0.5 | 1.0 |
| SLT (ours) | **24.2±0.2** | **31.2±0.3** | **34.6±0.1** | | **21.7±0.9** | **29.7±0.3** | **31.8±0.4** | |
| Small model | 8.9±0.2 | 18.4±0.0 | 30.3±0.1 | 41.6±0.5 | 8.4±0.3 | 16.2±0.2 | 27.3±0.3 | 38.7±0.3 |
| FedRolex [20] | 3.4±0.2 | 11.4±0.3 | 21.3±0.5 | | 2.8±1.1 | 10.2±0.9 | 18.4±0.4 | |
| FD [9] | 0.3±0.2 | 6.2±0.2 | 16.8±0.5 | | 0.1±0.0 | 0.1±0.0 | 15.7±0.4 | |

- **Measurement of activations and gradients**: To measure the size of the activations that have to be kept in memory, as well as the gradients, we apply `backward_hooks` to all relevant PyTorch modules in an NN. Specifically, we add these hooks to `Conv2d`, `BatchNorm2d`, `ReLU`, `Linear`, and `Add` operations. If a hook attached to a module is called, we add the respective size of the activation map and the size of the calculated gradient to a global variable to add up all required activations and gradients.

