# OpenReview forum: "Aggregating Capacity in FL through Successive Layer Training for Computationally-Constrained Devices"
_NeurIPS.cc/2023/Conference — NeurIPS 2023 poster_

### Official Review · Reviewer_akhJ · 2023-06-30

**Soundness:** 3 good
**Presentation:** 4 excellent
**Contribution:** 3 good
**Rating:** 7
**Confidence:** 4

**Summary:**

The paper focuses on enabling federated learning (FL) among clients that are bounded by the memory capacity. To this end, the work proposes successive layer training (SLT), a technique that sustains the memory usage below a given memory budget throughout the FL process. SLT partitions the model into three parts: *i)* a first part with frozen weights, *ii)* a intermediate part where all weights are trained, and *iii)* a final part (which can stop before the end of the architecture) whose only a fraction of the weights are trained, controlled by a parameter *s* which selects the number of channels to be trained. At training time, the FL process is broken into *N* stages. At each stage, the layers that are included in each of the three parts are changed. Progressively, the final part is moved towards the last layers of the network, until the whole model has been trained. The proposed scheme is applied on CNNs and compared with a number of baselines that employ nested submodels.

**Strengths:**

1) The proposed STL technique is interesting. Its parametrisation with respect to the indices of three-part network partitioning and the number of configuration steps provides a high level of flexibility and can thus be adapted to various settings, *e.g.* different memory constraints and clients with significant heterogeneity in their memory capacity.
2) The paper is well written, the proposed method is clearly described and positioned within the existing FL approaches.
3) The evaluation includes appropriate baselines and covers adequately different setups.

**Weaknesses:**

1) Althought STL seems to outperform the evaluated baselines, the potential reasons behind this performance gain is not discussed. For example, in the Heterogeneous Memory Constraints setup (Section 3.4), despite being the most realistic, the paper does not discuss why STL performs better than HeteroFl and FjORD, *i.e.* the two strongest baselines for this setup. The *Heterogeneity results* paragraph describes the results of Table 3 in a textual form, but does not attempt to explain why these are as they are.
2) The paper does neither discuss the selection of parameter *N*, which determines the number of distinct configurations to be used during the FL process, nor experimentally evaluates its impact. Similarly, there is no investigation on how these configuration steps are distributed across the global rounds. This is briefly touched upon in the last paragraph of Section 3.2, but this is not sufficient. More thorough investigation is needed, for STL to be useful. The key question is how to train using STL, *i.e.* how many configuration steps to use and how much time to spent using each configuration.

**Questions:**

1) Please specify the number of configuration steps *N* and the duration of each step for the presented experiments.
2) See the two points in Weaknesses.
3) As the work focuses only on memory constraints and not computational capabilities, it would be better to modify the title to reflect that, *e.g.* "Aggregating Capacity in FL through Successive Layer Training for Memory-Constrained Devices" instead of "Computationally Constrained".


**Limitations:**

The paper adequately covers its limitations and broader impact. All the main points are discussed.

---

> ### Author Rebuttal · Authors · 2023-08-09
>
> We like to thank the reviewer for the fair and constructive feedback on our manuscript. We will consider changing the papers title.
>
> # Performance of HeteroFL and FjORD
> We think the performance of HeteroFL and FjORD (Section 3.4) is limited due to the same effect that is present in FedRolex and FD: the limited co-adaptation that is possible within a layer, and calculating gradients of a subset of the layer without considering the remaining layer weights that reside on the server model. We show in Appendix A that the more distinct subsets are used the lower the accuracy gets, and that even with 4-5 subsets, the accuracy in CIFAR10 can reduce from $80\%$ to $60\%$.
>
> HeteroFL and FjORD use in principle the same subset mechanism as FedRolex and FD. Specifically, to lower the resources, constrained devices train a fixed smaller subset that is merged on the server with the remaining parameters. For example in HeteroFL increasing the number of heterogeneity levels from three ($[0.125, 0.25, 0.5]$) to four ($[0.125, 0.25, 0.5, 1.0]$) reduces the accuracy from $24.1$p.p. to $23.3$p.p., despite $\frac{1}{4}$ of devices training the full NN in the latter case. In SLT however, we avoid that devices train different subsets of a layer. We ensure that each device (independent from resources) trains the same head $F_H$, while allowing stronger devices to freeze fewer layers.
>
> # Number of steps $N$ for different NN models and memory constraints
> The number of steps $N$ depends on the choice of $K_F$ and $K_T$. For the full argument of why we select $K_T = K_F + 1$, please refer to the answer to reviewer tzDU. As a consequence of this design choice, only a single layer gets *filled up* per step. Therefore, in general, we require as many steps as there are layers in an NN. However, if the memory limit allows within a step to apply $s=1$, all remaining layers can be trained in full width, thus no further step is needed. Consequently, the number of steps also depends on the given memory constraint.
>
> The following table gives the number of steps for each NN architecture and evaluated constraints.
> | constraint $s$ | ResNet20 | ResNet32 | DenseNet40 |
> |---|---|---|---|
> | 0.66 | - | - | 10 |
> | 0.5 | 8 | 14 | - |
> | 0.33 | - | - | 15 |
> | 0.25 | 12 | 26 | - |
> | 0.125 | 16 | 14 | - |
>
> # Distribution of $N$ steps to $R$ rounds
> The following describes our methodology to distribute $N$ steps to $R$ rounds. The mapping of $N$ to $R$ for four of our experiments is visualized in **Figure R.1** in the rebuttal pdf. Generally, a layer that is trained should receive sufficient amount of training, to extract useful features for the downstream layers, but at the same time, it should not overfit in the current configuration. Hence, there is a tradeoff between too little training and too much training within a step.
> We distribute all steps $N$ over total rounds $R$ by calculating the share of parameters that are added to the training when transitioning from step $n$ to $n+1$. We calculate the number of trained parameters in the NN, based on $K_F$, $K_T$, and $s$ using $Q$, s.t.
> $Q(K_F, K_T, s) = \bigg( \sum_{k \in \\{k:K_F < k \leq K_T\\}}  P_k M_k \bigg) + P_{K_T+1}\lfloor sM_{K_T+1}\rfloor + \sum_{k \in \\{k:K_T+1 < k \leq K\\}}\lfloor sP_k\rfloor \lfloor sM_k \rfloor$
> Using $Q$ we calculate the number of rounds per step $R_N$,
> $R_n = R \frac{Q(K_F^{(n)},K_T^{(n)},s^{(n)})}{Q(0,0,1)}$ in case $n = 0$,
> $R_n = R \frac{Q(K_F^{(n)},K_T^{(n)},s^{(n)}) - Q(K_F^{(n-1)},K_T^{(n-1)},s^{(n-1)})}{Q(0,0,1)}$ else
> and lastly, the switching points $r_n$ by using
> $r_n = R_n + \sum_{i=0}^{n-1} R_j$.
>
> We observe that distributing the rounds based on the number of added parameters produces robust results throughout all experiments with fast convergence and high accuracy. We have also evaluated other empirical approaches like sharing rounds equally among rounds, which resulted in slower convergence and did result in lower final accuracy (e.g., $-3\text{p.p.}$ for CIFAR10). Additionally, we considered dynamic approaches, that switch based on the test accuracy/loss on the server (a similar technique has been used by Kundu and Jaja [1]). However, such approaches require dataset-specific hyperparameter tuning, and our results show that they do not produce necessary better results, as shown in **Figure R.4** in the rebuttal pdf.
>
>
>
> [1] Amit Kumar Kundu and Joseph Jaja. "FedNet2Net: Saving Communication and Computations in Federated Learning with Model Growing." International Conference on Artificial Neural Networks. Cham: Springer Nature Switzerland, 2022.

---

> > ### Comment · Reviewer_akhJ · 2023-08-14
> >
> > Thanks for the precise answers. Please make sure to integrate the responses to the main part of the paper.

---

### Official Review · Reviewer_tzDU · 2023-07-02

**Soundness:** 3 good
**Presentation:** 3 good
**Contribution:** 2 fair
**Rating:** 5
**Confidence:** 3

**Summary:**

The paper proposes an FL training methodology to reduce the memory footprint for constrained devices. Specifically, the layers inside the original model are divided into three categories, i.e., frozen, fully trained, and partially trained. The frozen layers only participate in the forward pass but do not require storing the activations or computing gradients for the backward pass. The partially trained layers also only train a fraction of the parameters, thus reducing the memory overhead. The authors further provide heuristics for determining the configuration of layers within these three categories throughout training and the number of training steps allocated to each configuration.

**Strengths:**

- The paper is well-written, easy to follow, and the ideas are clearly presented.
- The experiments investigate the method from different aspects like iid versus non-iid data distribution as well as iid versus non-iid compute devices, and show benefits over several baselines.

**Weaknesses:**

- While the proposed method works well on the evaluated benchmarks, it appears to be a straight-forward combination of existing ideas (using s ratio of parameters to partially train layers + progressive freezing). As such, the reviewer finds the originality of the ideas rather limited.
- Can the authors provide some intuition or theoretical justification for acceptable bounds on s and the general configuration of F_F, F_T, and F_H? Specifically, what is the bound on these design hyperparameters after which the convergence is impeded?
- There seems to be a large gap between the performance of prior work (FedRolex) versus the performance presented in the original paper. Can the authors clarify any difference in the experimental setup that may have led to this?
- It would be beneficial to present the corresponding training memory footprint (activation+parameters+grads) for each benchmark to better clarify what constraints each benchmarked s_fd ratio corresponds to, and see e.g., how aggressive is 0.125 ratio.

**Questions:**

Please see weaknesses above.

**Limitations:**

The authors do not discuss the potential negative societal impact of the work in the main text.

---

> ### Author Rebuttal · Authors · 2023-08-09
>
> We like to thank the reviewer for the fair and constructive feedback on our manuscript.
>
> # SLT design choices ($K_T, K_F, s$)
> Our general choice of setting $K_T = K_F + 1$ while maximizing $s$ throughout all configurations $N$ is motivated by preliminary ablation studies we conducted (please refer to **Figure R.3** in the rebuttal pdf). Specifically, we investigated, given a specific memory budget, should $K_T$ or $s$ be maximized, as increasing both increases the memory consumption in training. We evaluated the accuracy in a centralized training setup (no freezing applied, i.e. $K_F=0$). The results suggest that maximizing $s$ in favor of $K_T$ clearly results in higher accuracies, therefore, we only *fill up* and train a single layer per step (i.e. the minimal possible number of layers). $F_H$ is given by the design choice of $K_T$.
>
> # SLT Memory limit / minimal $s$
> The memory reduction limit of SLT is in general affected by two things:
> * Firstly, the limit is constrained by how much memory it requires to *fill up* and train a single layer of the NN (relative to full training). Note that the memory cost of training a single layer compared to the training of the full NN decreases as we have more layers.
> * Secondly, it is influenced by how low $s$ can be set. Note that the subset of a layer of the head $w$ is defined by $s$, ($w \in \mathbb R^{\lfloor sP \rfloor \times \lfloor sM \rfloor}$). Hence, the minimum of $s$ would be determined by $P$ and $M$, since at least a single filter per layer has to be trained. Consequently, if a layer only has 6 output filters, the minimum applicable $s = \frac{1}{6}$.
> SLT allows for memory reduction of $0.125\times$ for ResNet20, $0.09\times$ for ResNet32, and $0.08\times$ for ResNet56.
>
> # Accuracy difference between our implementation of FedRolex and the FedRolex paper
> We are fairly confident that our implementation of FedRolex reflects what is described in the original paper since we used the paper and the authors' public source code for implementation. We additionally ran our implementation against theirs and confirmed that throughout the rounds the same subset of parameters get modified on the FL device. We think the differences in the FedRolex paper (Figure 4) are caused by the following aspects:
>
> * In FedRolex, the authors use a significantly larger NN model to evaluate CIFAR10 (generic ResNet18, $11$M parameters) while we use the specifically adapted version for CIFAR10 (ResNet20, $0.27$M parameters).
>
> * FedRolex scales down the NN with $\gamma=\{2,4,8,16\}$, where $\gamma$ refers to **size** (parameters) reduction of the global model, while we evaluate the reduction of memory that is mainly dominated by the activations. As a consequence, when scaling down the NN with $s$, the size of the NN reduces quadratically, while memory only reduces linearly.
>
> To demonstrate this, we added $\gamma$ and memory scaling into a table, comparing the accuracy results from Figure 3 of the FedRolex paper and our results from Figure 1.
> | $\gamma$ | $1$ | $2$ | $4$ | $8$ | $16$ |
> |---|---|---|---|---|---|
> | memory reduction | $1$ | $1.41$ | $2$ | $2.82$ | $4$ |
> | FedRolex (paper, Figure4) | $84.5\\%$ | $68\\%$ | $58\\%$ | $58\\%$ | $58\\%$ |
> | FedRolex (ours, Figure1) | $87.6\\%$ | - | $80\\%$ | - | $61\\%$ |
>
> In our experiment (Figure 1), we evaluate with IID data while FedRolex evaluates "low heterogeneity" of data. The remaining differences are the number of rounds (2000 vs. 2500), learning rate, and learning rate decay schedules. We use sinus-decay, while FedRolex uses fixed steps to lower the learning rate. When comparing full training (i.e. $1\times$) this already creates an accuracy improvement of ~$3$p.p. for FedRolex. So in practice, our setup allows for (slightly) higher final accuracies.
>
> Apart from these, the setup is fairly similar using 100 devices with 10 active per round. We, therefore, think the differences in accuracy mostly come down to the choice of the NN model and the distribution of data.
>
> # Memory footprint of models and experiments
> Generally, the maximum amount of memory for training is dominated by the activations. With scaling down the number of filters using $s$, the activation maps linearly reduce. The number of parameters and respective gradients however reduce quadratically. The following tables show the maximum memory consumption during training and the respective contributions of activations, weights, and gradients for end-to-end training of the NN and scaled-down models (small model).
>
> |**Full NN training** |ResNet20|ResNet32|DenseNet40|
> |---|---|---|---|
> |activations (MB)|$96.0$| $853.8$| $408.7$|
> |weights (MB)|$1.2$|$2.8$|$0.7$|
> |gradients (MB)|$1.2$|$2.8$|$0.7$|
>
> |**Reduction factor $s=0.125$** |ResNet20|DenseNet32|DenseNet40|
> |---|---|---|---|
> activations (MB)| $12.0$ | $106.7$|   $50.3$
> weights (MB)| $0.02$| $0.05$ |  $0.01$
> gradients (MB)| $0.02$ | $0.05$ |  $0.01$|

---

> > ### Comment · Reviewer_tzDU · 2023-08-21
> > **Post-rebuttal feedback**
> >
> > Thank you for your rebuttal. I have adjusted my score accordingly.

---

### Official Review · Reviewer_AkhZ · 2023-07-06

**Soundness:** 3 good
**Presentation:** 3 good
**Contribution:** 3 good
**Rating:** 6
**Confidence:** 4

**Summary:**

This paper introduces an approach known as Successive Layer Training (SLT) for federated learning (FL) on edge devices with limited resources. The core concept of SLT involves training a portion of the neural network (NN) parameters on the devices while keeping the remaining parameters fixed. Subsequently, the training process progressively expands to include larger subsets until all parameters have been trained. The authors assert that SLT offers several advantages, including reduced memory requirements for training, enhanced accuracy and convergence speed in FL, and compatibility with heterogeneous devices that possess varying memory constraints. To validate their claims, the authors conduct evaluations of SLT on multiple datasets and NN architectures. Additionally, they compare SLT against state-of-the-art techniques such as Federated Dropout (FD), FedRolex, HeteroFL, and FjORD.

**Strengths:**

1, This paper is well-motivated. Memory footprint is a key constraint in training models on edge devices. The proposed approach allows for co-adaptation between parameters and reduces the memory footprint.
2, The paper is well-written, offering clarity and organization.
3, This paper conducted a comprehensive experimental analysis on four datasets (CIFAR10, CIFAR100, FEMNIST, and TinyImageNet) and three NN architectures (ResNet20, ResNet32, and DenseNet40). Comparisons are made with several baselines, and the results demonstrate that SLT achieves higher accuracy and faster convergence in both independent and non-independent and identically distributed (iid and non-iid) settings. Additionally, the performance of SLT is investigated in heterogeneous environments, where devices have different memory constraints.


**Weaknesses:**

The paper primarily focuses on addressing memory constraints in federated learning (FL) by introducing the Successive Layer Training (SLT) technique. However, it does not thoroughly consider other types of resource constraints, such as computation or communication, which are also crucial aspects of FL on devices. The paper does not provide insights into how SLT impacts computation time or communication costs in the FL process. It is important to acknowledge that SLT may introduce certain overheads or inefficiencies in these areas, potentially influencing its practical applicability and overall performance. Further investigation and analysis of the computational and communication implications of SLT would provide a more comprehensive understanding of its practicality and effectiveness in resource-constrained scenarios.

**Questions:**

In the evaluation, the used model sizes are relatively large and can be over-parameterized for the dataset. As a result, the method Small Model, which is still sufficient for the task, can achieve better performance than the SOTA methods. If the model size goes smaller (i.e., LeNet with 1x, 2x, or 4x channels), do the proposed method and the method Small Model still outperform the SOTA baselines?

**Limitations:**

Please see the weakness and questions.

---

> ### Author Rebuttal · Authors · 2023-08-09
>
> We like to thank the reviewer for the fair and constructive feedback on our manuscript.
>
> # Model capacity
> ## Model capacity in conducted experiments
> The observation is correct that in some experiments the NN architecture is over-parametrized for the given dataset. We observe this, especially for the FEMNIST dataset from the Leaf benchmark. Here, using a memory constraint of $0.5\times$, only reduces the accuracy from $87.6\\%$ to $86.9\\%$. However, we observe that if the dataset is more complex and capacity is not sufficient, SLT shows larger gains over a small model (e.g., when using tinyimagenet and ResNet32).
> ## Effectiveness of SLT and baselines with small models
> In general, we observe that SLT can reduce the memory requirement (relative to the full NN) more effectively if the NNs have more layers, as the minimum memory SLT requires for training is mainly dominated by the memory cost of *filling up* a single layer. Hence, when having only a few layers, as it is the case with LeNet, SLT is not as effective w.r.t. memory reduction, and hence we do not expect SLT to perform much better than a small model. Additionally, we confirmed that the trends of the gap between a small model and FedRolex still exist, even when capacity is sufficiently lower. We Evaluated a minimal ResNet structure with only 9 layers and scaled-down memory using $0.125\times$. We observe that the small model reaches an accuracy of $57.7\\%$, while FedRolex only reaches $25.6\\%$.
>
> # Communication and computation in SLT
> We differentiate between memory constraints and constraints on communication and computation (FLOPs), as the latter are less restrictive constraints (e.g. referred to as soft constraints in [1]). Specifically, insufficient available memory excludes a device from the training (as it is done with Google GBoard training [2]), while constraints on communication and FLOPs can slow down the convergence (e.g., w.r.t. time or energy). Hence, when dealing with communication and FLOPs resources, one should opt for using them efficiently.
>
> ### Communication efficiency of SLT
> We want to point out that we already consider communication efficiency in our evaluation (Figure 4). We show (Figure 4: x-axis is the uploaded volume of the data to the server, and hence the communication overhead) that we require significantly less communication (GBytes) to reach the same accuracy as state of the art. Also, with respect to the communication overhead, we converge as fast as a small model but enable higher final accuracies. This is in part due to the freezing of layers, as the parameters of frozen layers do not change, and hence do not need to be uploaded to the server, increasing the communication efficiency.
>
> ### Computation (FLOPs) Efficiency of SLT
> We performed a similar evaluation to show the FLOPs overhead in federated training for SLT and the related baselines. We observe similar trends: 1) we require significantly fewer FLOPs to reach a certain level of accuracy, compared with state of the art; and 2) the number of FLOPs required by SLT to reach a certain accuracy is similar to the small model, but SLT enables higher final accuracies. The results are shown in **Figure R.2** in the rebuttal pdf.
>
> We would add these results to a revised version of the manuscript.
>
> [1] K. Pfeiffer, M. Rapp, R. Khalili, J. Henkel, "Federated Learning for Computationally-Constrained Heterogeneous Devices: A Survey", ACM Computing Surveys, Volume 55, Issue 14, 2023
> [2] Timothy Yang, Galen Andrew, Hubert Eichner, Haicheng Sun, Wei Li, Nicholas Kong, Daniel
> Ramage, and Françoise Beaufays. Applied federated learning: Improving google keyboard query suggestions. arXiv preprint arXiv:1812.02903, 2018.

---

### Official Review · Reviewer_iW6b · 2023-07-16

**Soundness:** 3 good
**Presentation:** 2 fair
**Contribution:** 2 fair
**Rating:** 6
**Confidence:** 4

**Summary:**

In this paper, authors study the case where the edge device is not capable of holding the training of the whole network due to limited memory. Authors find that it is necessary to train a whole network rather than train a submodule as previous methods do. Motivated by this observation, authors propose to first scale down the intermediate size of the network and then expanding it as its precedent layers being fully trained. Authors provided experimental results and those results successfully proved that SLT outperforms other algorithms in this scenario.

**Strengths:**

The observation that it is necessary to train a full network, rather than a submodule, sounds reasonable to me. Moreover, the SLT method proposed looks new to me. This idea could be helpful for future studies in memory-constrained FL training.

**Weaknesses:**

The paper is not clearly written in some parts, such as Section 2.2. My major concerns come from two parts:
1) Why SLT can achieve almost the same result w.r.t. the ''small model'' method when the model can accommodate the edge device, as shown in Figure 4? To me, the shallower layers are only trained when the deeper layers get forward partially in the early stage, and they are kept frozen when training the deeper layers. This means the shallower layers are not fully trained with the correct gradient that comes from the whole model, which means SLT is not equivalent to the full training algorithm and it shall get a worse performance than the ''small model'' method. Therefore I doubt why it can get a matched accuracy as shown in Figure4.
2) Actually the footprint of activations can be alleviated by checkpointing or offload, please compare SLT with these methods.

**Questions:**

Please refer to my questions in the weakness part.

**Limitations:**

Authors have clarified their limitations.

---

> ### Author Rebuttal · Authors · 2023-08-09
>
> We like to thank the reviewer for the fair and constructive feedback on our manuscript.
>
> # SLT's improvements over a "small model" baseline
> * The observation is correct that parts of the model are never trained end-to-end in SLT. However, SLT pretrains a subset of all layers end-to-end in step $0$ (equal to an affordable small model), where no layer is frozen $K_T=0,K_F=0$. After this pretraining phase, gradually, step by step, a layer is *filled up* while shallower layers get frozen. This way shallower layers are trained to still produce useful features for the downstream head. Thereby, SLT reaches higher accuracies than a small model that has the same memory footprint in training. This is especially the case if the small model does not have sufficient capacity for the given dataset. SLT allows training models with larger capacities. Despite the error introduced by our training scheme being not end-to-end, the gain in capacity outweighs this error as can be seen in Figure 4 and Table 1. At the same time, SLT is not able to reach the accuracy of the full NN, hence can not compete against an end-to-end trained full NN.

---

> > ### Comment · Reviewer_iW6b · 2023-08-21
> >
> > Thank you for your rebuttal. I have adjusted my score accordingly.

---

### Official Review · Reviewer_cUHB · 2023-07-28

**Soundness:** 3 good
**Presentation:** 3 good
**Contribution:** 3 good
**Rating:** 5
**Confidence:** 3

**Summary:**

This paper proposed a new training method for neural networks in the federated setting, called successive layer training. This method tries to train NN layer by layer by freezing the trained layer and scaling down the NN heads. The authors have a clear presentation for this method and propose detailed experiments to show the performance.

**Strengths:**

The experimental results show good performance when compared with other algorithms for this memory-constrained setting. It is exciting to see that such an (approximate) layer-by-layer training strategy can have an acceptable accuracy. The configuration for this setting is also flexible when adapting to environments with different resource constrained.

**Weaknesses:**

The method itself is somewhat straightforward and the paper makes no explanation about why this strategy can work. Meanwhile, the experiments were only done on some toy neural networks and thus were not very representative. At least testing on Resnet50 on Imagenet is necessary.

**Questions:**

Although this strategy here is proposed under the background of FL, it is a general approach suitable for any distributed environment. So is it effective for applying the method in standard distributed training, while communication is the bottleneck.  Meanwhile,  is it possible to apply this strategy to attain the accuracy of full training?

**Limitations:**

Some suggestions are as follows:

1. Give some explanation for this method
2. Do adequate experiments on standard nets and datasets
3. Study this training strategy as a general approach first and then apply it to FL

---

> ### Author Rebuttal · Authors · 2023-08-09
>
> We like to thank the reviewer for the fair and constructive feedback on our manuscript.
>
> # Scope of the work
> Our work is motivated by the observation that the current state of the art in memory-aware-FL through scaling of the NN does not work well. We think that the bad performance is caused by the fact that gradients of the subset of a layer are calculated (for a local epoch) without consideration of the parameters that reside on the full server NN (as also discussed in Appendix A). We think this problem is tightly coupled to the federated averaging mechanism which is used in federated learning. Hence, as the state of the art does, we target federated learning environments.
>
> We show in Figure 4 in the paper that SLT requires significantly less communication to reach the same level of accuracy as the state of the art, so SLT could be also effective in distributed training settings where the communication is the bottleneck. However, this should be studied more in detail and is thus delegated to future work.
>
> # ResNet50/ImageNet
> We think that currently for devices in federated learning, such as smartphones, IoT devices, and sensors, it is not feasible to train large models like ResNet50 on ImageNet (full image resolution). We want to add that we also lack the capabilities hardware-wise to simulate ResNet50/ImageNet in a federated environment as a simulation of FL takes considerably more time than centralized training.
>
> Through our study, we observed that SLT benefits from two factors: a) a larger capacity gap between a small model ($0.125\times$) and the full model ($1.0\times$); and b) deeper NNs, as the memory cost of *filling up* a single layer reduces, thus allows for larger values of $s$ throughout the training. We believe that this effect would also transfer to larger NNs and more complex problems like ResNet50 and ImageNet.
>
> To provide further evidence for that we evaluated a downscaled version of ImageNet (64 by 64 pixels) with ResNet56 (one of the cheaper CIFAR10 adapted family of ResNet models), where with 500 devices SLT reaches $22.0\\%$ of Top-1 accuracy, while the small model baseline only reaches $8.2\\%$ Top-1. It can be seen that compared to TinyImageNet, this is already a larger gap. We are confident that the FedRolex and FjORD would perform worse than a small model, as it is the case for all other experiments.

---

> ### Comment · Area_Chair_1KKn · 2023-08-14
> **Does the proposed approach admit theoretical guarantees in convergence and accuracy**
>
> Wondering if the proposed approach admits theoretical guarantees in convergence and accuracy. It looks like an applicatoin of coordinate descent under the stochastic setting.

---

> ### Author Response · Authors · 2023-08-14
> **Reply to AC (1KKn) comment**
>
> We want to thank the area chair for this comment and the interesting observation. We agree that parts of our technique can be interpreted as (block) coordinate descent, as we freeze layers (i.e. fixing these variables) while optimizing the remaining ones. However, we also see some differences to coordinate descent:
>
> For the sake of simplicity, we ignore the stochastic part. Also, let $w$ be the set of full parameters, $w_x$, $w_y$ be subsets of $w$, $l(w)$ be the loss w.r.t $w$, and $g_w$ be the gradients w.r.t $w$.
>
> In (block) coordinate descent (in a cyclic step), a gradient for a subset of parameters $g_{w_x}$ is calculated based on the loss surface of all parameters $l(w)$. This is different from our approach, as in SLT, we first calculate gradients for a subset of the parameters ($g_{w_x}$) based on the loss surface of the subset $l(w_x)$. We then freeze parameters (i.e. early layers) while adding remaining parameters to the head followed by essentially coordinate descent, i.e. we calculate and apply $g_{w_y}$ based on $l(w_x \cup w_y)$. Throughout SLT's training steps, the process of adding parameters and freezing layers is repeated.
>
> In essence: in coordinate descent, the optimization problem stays the same throughout the training, while in our case the loss surface $l()$ changes as more parameters are added.
> We, therefore, think that existing convergence guarantees for coordinate descent can not be applied to our technique.
>
> Throughout our empirical evaluation, we also made the following observations w.r.t. accuracy:
>
> * SLT reaches higher accuracy than FedRolex and FD, that both essentially, in order to eventually train all parameters $w$, cycle between calculating and applying gradients $g_{w_x}$ based on $l(w_x)$ followed by calculating and applying $g_{w_y}$ based on $l(w_y)$, hence, changing the loss surface constantly. In SLT however, it is ensured that when calculating $g_{w_y}$, the loss surface of all *already trained* parameters is $f(w_x \cup w_y)$ is considered.
> * FedRolex and FD perform worse than only doing $g_{w_x}$ based on $l(w_x)$ throughout the training, as is the case with a small model baseline.
> * Given a constraint on memory, SLT cannot reach the same accuracy as end-to-end training without any constraints.

---

### Author Rebuttal · Authors · 2023-08-09

We attached a pdf with four figures (R.1, R.2, R.3, R.4) to support our answers to the reviewers' questions.

---

### Decision · Program_Chairs · 2023-09-21

**Decision:**

Accept (poster)

**Comment:**

This paper considered the memory restricted scenario while training a large DL model. It proposed a block coordinate descent alike algorithm to solve such challenge. This paper has included extensive experiments to justify the proposed approach. All reviewers are positive to accept this paper. The following aspects can be polished in the final version:

- provide explanation to show the convergence and correctness of the proposed approach.
- add the makeup experiments in this paper
- also discuss the computation and communication restricted scenarios.